# How Do We Keep Our New Graduate Nurses in Australia?

**DOI:** 10.3390/nursrep15080276

**Published:** 2025-07-30

**Authors:** Linda Ng, Rob Eley, Jennifer Dawson, Priya Govindaswamy, Karen Walker

**Affiliations:** 1School of Nursing & Midwifery, University of Southern Queensland, Ipswich, QLD 4305, Australia; 2Faculty of Medicine, The University of Queensland, Brisbane, QLD 4072, Australia; r.eley@uq.edu.au; 3Royal Women’s Hospital, Melbourne, VIC 3050, Australia; jennifer.dawson3050@gmail.com; 4The Sydney Children’s Hospital Network, Sydney, NSW 2145, Australia; priya.govindaswamy@health.nsw.gov.au; 5Prince Alfred Hospital, University of Sydney, Sydney, NSW 2050, Australia; karenwalker050@gmail.com

**Keywords:** millennial nurses, new graduate nurses, nursing, retention, shortages, theory–practice gap, workplace culture

## Abstract

This paper aims to discuss the transition of new graduate nurses into the workforce, the preparation provided to equip them through the novice–beginner stage, and the theory–practice conundrum. **Background**: In Australia, new graduate transition programs have been in existence since the 1990s. While there is widespread acknowledgment that this period is pivotal for new graduate nurses entering the profession, there is a lack of consensus on the definition of best practice to achieve optimal preparation for new graduate nurses transitioning into the workforce. **Methods**: This discussion paper integrates the nursing literature on this topic with the extensive professional experiences of the authors, who are currently working as clinicians in metropolitan hospitals and hold academic positions at universities. Their insights are informed by the literature sourced from peer-reviewed English language journals, including reviews, empirical studies, and national and international reports. **Discussion:** Recruiting and retaining nurses presents a multifaceted challenge that requires the development of effective tools and strategies to build a sustainable workforce. Both the literature and the authors’ experiences highlight several key factors influencing the preparedness of new graduates. These factors include workplace culture, the demands placed on new graduates, and the support, education, and training they receive. The perspectives shared in this article offer valuable discussion points that can deepen our understanding of the current issues and contribute to the development of more effective solutions.

## 1. Introduction

On a global scale, the nursing workforce is growing older, and there are fewer young individuals entering the profession to replace the substantial number of baby boomers who are on the verge of retirement in the next decade [1]. Furthermore, dissatisfaction among nurses is prevalent, resulting in sub-optimal retention. Factors such as heavy workloads [2], shift work [3], inadequate mentoring, supervision, and support [4], low wages [5], poor working conditions [6], limited professional/career advancement [7], and restricted autonomy [8] are cited as contributing to the early departure of nurses from the sector.

Simultaneously with shortages to replace the current workforce, the demand for healthcare services is on the rise, driven largely by an ageing population with multiple chronic conditions. As a result, there is a need for more nurses to meet the expanding healthcare requirements. The COVID-19 pandemic has further complicated staffing issues and created additional stress in healthcare settings [9]. This heightened stress is causing reduced job satisfaction [10], increased rates of compassion fatigue [11], and burnout [12]. This combination of factors is leading nurses from new graduates to experienced staff leaving the profession, particularly in acute care settings [13].

In 2021, the Australian Government’s Care Workforce Labour Market Study [14] anticipated a deficit of over 200,000 full-time care workers in Australia by 2050. The Committee for Economic Development of Australia [15] foresees an even greater shortage. They predict a potential shortage of 110,000 or more workers in the aged care sector alone in the next decade, with a staggering estimate of over 400,000 by 2050. However, it is not just the nursing workforce that is affected; the situation is health sector-wide and global. The World Health Organization [16] projects a global shortage of 10 million healthcare workers by 2030. Collectively, these projections paint a grim and potentially catastrophic picture of supply in future healthcare workforce if decision-makers do not find solutions to make the sector more attractive to current, prospective, and new employees.

This opinion piece begins by discussing the systemic and structural challenges contributing to the current and projected nursing workforce shortages, with a particular focus on the Australian context. It then offers a series of recommendations aimed at improving workforce sustainability, retention, and attraction, drawing on both national and international evidence to inform practical and policy-level solutions.

## 2. Discussion

### 2.1. Age and Retirement in the Nursing Workforce

The ageing workforce and the retirement of experienced nurses are critical issues that strip the healthcare sector of valuable clinical expertise. This trend not only disrupts the balance within the workforce but also results in a significant loss of mentorship for incoming staff. Baby boomers, born between 1945 and 1964, are now at or nearing retirement age. In 2022, the average age of nurses in Australia was 43 years, with 46.8% of nurses aged 45 or older and 11.6% over 60 years old [17]. The United Kingdom (UK) faces a similar challenge, with one in five nurses being over 50 [18]. This pattern is evident globally, including in the United States (US), New Zealand, Canada, and various European countries, all of which report an ageing nursing workforce [18]. If this trend continues, it is projected that within the next two decades, approximately 75% of the current nursing workforce in these countries will retire.

Adding to the predicament of an ageing workforce is the age at which undergraduate nursing students begin and complete their courses. Historically, as older nurses retired, they were replaced by younger ones. New nurses are pivotal to ensuring older generations are replaced and the balance in the nursing workforce is maintained. However, nursing seems to have lost some of its appeal as a profession, particularly for young people [19]. Evidence suggests that globally, fewer young school leavers, typically aged around 17 or 18, who used to be the mainstay of undergraduate nursing programs, are opting for nursing as a career [19].

Losses in the workforce are not restricted to those nurses of retirement age. Nurse resignations also exhibit generational variations. The UK Nursing and Midwifery Council reported that 52% of individuals who deregistered in 2022 did so before their expected departure, and according to the Royal College of Nursing, nearly 43,000 nursing and midwifery professionals aged 21 to 50 exited the field between 2018 and 2022 [20].

Millennial nurses, those born between 1981 and 1996, are projected to become the largest age group in the nursing workforce by 2025 [19]. Yet these nurses are also leaving acute care settings and the nursing profession at higher rates [13], and as a result, nurse managers and organisations need to implement a targeted retention strategy for them.

In 2025, the largest cohort of new graduates replacing retiring and departing nurses would be expected from Generation Z, born 1997–2012. However, as highlighted in a recent review, the number of new graduate nurses who leave nursing and express no intent to return is substantial [21]. Previous research has established that between 18% and 30% of new graduate nurses will leave their current position for a different practice environment or leave the profession altogether in the first year and between 37 and 57% will leave in their second year of practice [22].

This shift from the historical cycle, where younger nurses joined the workforce as older ones retired and retention rates were high, threatens to create an imbalance in the nursing workforce. This imbalance has substantial implications not only for future recruitment but also for retention. Improving retention rates will be crucial for efficiently addressing the ongoing nursing workforce crisis, ensuring an adequate number of nurses to replace the “baby boomer” generation as they retire over the coming decade or two.

Nursing students pursuing their undergraduate degrees and recent nursing graduates symbolise the future nursing workforce. Any issues associated with students discontinuing their studies or new nursing graduates exiting the profession will only worsen the existing and anticipated nursing shortages. Optimal preparation of new nursing graduates into the workforce is thus of major importance.

The aim of this discussion paper is to highlight the major factors that are contributing to the decline in the nursing workforce and offer recommendations to make the profession more attractive to new nursing graduates. The paper discusses factors affecting that preparation and for the purpose of discussion, considers new graduates to consist of both younger millennials and those of Generation Z who are entering or about to enter the workforce. Both millennial and Gen Z nurses possess similar professional skills, learning and communication styles, and outlooks reflecting their exposure to current technology and social trends.

### 2.2. Theory—Practice Gap

Despite educational efforts to provide nursing students with the necessary information to navigate the challenges of transitioning to practice, it is well-documented that feelings of being under-prepared, overwhelmed, and anxious persist among many new graduate nurses and contribute to early departure [23].

The lack of preparedness among new graduate nurses during their transition into practice—largely attributed to the persistent theory–practice gap—has been extensively documented in the literature, with minimal progress observed over several decades [24]. In fact, the gap between the expectations of employers hiring new graduate nurses and the standards set by universities remains as significant an issue today [25] as it did nearly four decades ago. There is consistent evidence indicating a deficiency in the practice readiness of new graduate nurses for their professional roles, leading to a significant and problematic developmental delay between acquired knowledge as students and their required knowledge enter the workplace as graduate nurses [26]. Additionally, there are notable variations in expectations between educational institutions and practice stakeholders, often influenced by the historical and social context within which nursing education and professional practice are situated [27].

Despite the anticipation that new nurses should possess entry-level competencies upon graduation, research reveals gaps in their knowledge, skills, and clinical judgment related to their graduate roles [28]. In the United States, a substantial proportion, ranging from 65% to 76% of new graduate nurses, did not meet the expectations for entry-level clinical judgment [29]. Similarly, Swedish new graduate nurses failed to meet expectations and were rated lowest in areas such as planning, prioritising, informing, and teaching co-workers and students, and they were involved in near misses, omissions, and errors in the execution of clinical skills [30]. Evidence from a synthesis of 45 reviews indicates persistent deficiencies in entry-level competencies, with new graduate nurses facing challenges in the six key areas of communication, leadership, conflict resolution, organisation and prioritisation, critical thinking and clinical reasoning, and stress management [31].

However, the same authors [31] concluded that major concerns about the practice readiness of new graduate nurses are not consistently supported by strong evidence. Despite this, they further conclude that there is strong evidence that new graduate nurses “lack confidence” during the first few months of employment.

Whether the deficiencies and lack of preparedness lie exclusively with skills or with confidence may be contextually specific and be a combination of both. Nevertheless, each point to the need for strong transitional support to supplement skills and to enhance confidence.

### 2.3. Transition Support

Healthcare organisations have a strong interest in retaining new graduate nurses [32]. Their primary challenge lies in effectively integrating these new graduate nurses into the demanding field of high-acuity nursing practice. Research consistently highlights the vulnerability of new graduate nurses in the early stages of their nursing careers, as they often experience feelings of uncertainty and isolation [23,26]. Without adequate support to address these challenges, it becomes exceedingly difficult for them to successfully transition from new graduate nurses to advanced beginner-level nurses and to seamlessly integrate into the system as effective team members [30]. The successful integration of new graduate nurses has a direct positive impact on patient safety, patient satisfaction, and staff retention, with a lack of engagement potentially leading to subpar performance and high turnover [28].

Transition programs for new graduate nurses, designed to facilitate their development and integration into the workforce, have been in existence since tertiary education became the norm. These programs are referred to in the literature using various terms such as transition to practice, nurse entry to practice, first year of clinical practice, residency, internship, new graduate nurse, and early career nursing programs [33,34]. Across different countries, professional, regulatory, and governmental bodies have not only recommended but also mandated such programs to address the challenges associated with the transition to professional practice in the workplace [33,34].

Consequently, there has been a notable increase in the establishment of transition programs, such as New Zealand’s National Nurse Entry to Practice programs [33] and the United Kingdom’s National Preceptorship Framework [35], which are now mandatory for all new graduate nurses in their respective countries. In the United States, transition programs are not mandatory, but since 2002, there has been a significant increase in the development of formal residency programs [36]. Similarly, in certain Canadian jurisdictions, provincial ministries of health have introduced funded supernumerary positions for new graduate nurses for up to six months to provide support during their transition [34].

The consensus in the literature on the transition of new graduate nurses emphasises the necessity for a supportive and non-threatening environment for the successful transfer of theoretical knowledge into practice [23]. In Australia, this belief has been implemented through the introduction of graduate transition programs, where new graduate nurses are supported in a safe learning environment in their first year of practice. It is widely acknowledged that these programs should occur within an organisational framework that effectively integrates new graduate nurses into the systems and processes of the organisations [32]. In line with this perspective, it is argued that transition programs should be designed to offer the appropriate level of support, fostering the development of confidence and competence among new graduate nurses [31].

There is considerable diversity in the elements of transition programs but typically, they consist of a mix of educational components, formal or informal preceptorships, mentorships, study days, supernumerary time, and orientation specific to the clinical unit. Extensive research exists on transition programs for new graduate nurses, revealing advantages for both the new graduate nurse and the employing organisation [1,24,31,36]. The findings suggest there is a benefit from incorporating considerable diversity in program elements, encompassing factors such as content, duration, the quantity and nature of clinical rotations, teaching and learning methods, and the incorporation of theory in program design. However, despite the considerable amount of research conducted in this field, there is a lack of consensus on what qualifies as best practice.

Similarly, while recommending the provision of support, the literature falls short of offering a precise definition of what qualifies as “support” [25,31]. This lack of clarity is particularly problematic for a comparison and benchmarking of practices and policy initiatives. Additionally, the literature does not include the significance and types of support for new graduate nurses, including the ideal timing and duration of support, the most suitable providers of support, the manner in which support should be delivered, and the types of barriers that hinder timely and appropriate support [1,23,25,32].

The concept of “support” for new graduate nurses is inherently fluid and context-dependent. What constitutes “support’ is not fixed; it is dynamic and must adapt to new developments in education and industry. It must evolve to meet the changing demands of the educational landscape, organisations, and societal expectations. It must also account for the perspective of new graduate nurses in designing adequate and effective support. As teaching methodologies, technological advancements, and industry standards evolve, so too must the strategies for supporting new graduates.

The local context significantly influences what constitutes “support”. Regional job markets, cultural expectations, and available resources vary widely, affecting the types of support that are most effective. Understanding and integrating these local nuances is crucial for developing relevant and effective support systems. Each university has its unique characteristics, including strengths, weaknesses, and educational philosophies. These factors shape the needs of their graduates and, consequently, the support they require. Tailoring support to the specific context of the graduating university ensures that it addresses the unique challenges and leverages the strengths of its graduates. Employers/organisations have diverse expectations and requirements, which must be considered when developing support strategies for new graduate nurses. Aligning support with the specific needs of employers can facilitate a smoother transition for new graduate nurses into the workforce. This alignment helps ensure that new graduate nurses are not only prepared for their roles but also able to meet the specific demands of their employers.

Whilst a description of programs may be found in the literature, data are limited as to the success of these programs. Reviews of studies of transition-to-practice interventions have concluded that the studies often have low methodological quality such as a lack of concrete indicators or theoretical foundation [37]. Nevertheless, there is some evidence of positive outcomes. For example, studies resulting from the New Zealand hospitals entry-to-practice programs showed they led to a “favourable effect on turnover” (p103), a “rise in job satisfaction among new graduate nurses” (p103), and an “80% increase in retention” of these nurses in the organisations [38]. Furthermore, Charette et al. [37] reported that a 12-month transition program at two teaching hospitals in Melbourne resulted in the new graduate nurses having greater confidence and competence. Despite these results, it is clear that more rigorous research is required to determine if transition programs effectively accomplish their objectives or if they are as successful as a prolonged period of practice in a supportive and stimulating clinical setting.

### 2.4. On-Going Support

Providing new graduate nurses with the support needed to perform their job is vital to their successful transition. There is no “one size fits all”. Given the diversity in the educational backgrounds, personal circumstances, and career aspirations of new graduate nurses, a single approach to support is unlikely to be effective for everyone. Customisation and flexibility are key to addressing the unique needs of each new graduate nurse. Support systems must be adaptable, allowing for adjustments based on individual needs and changing circumstances. Each program will reflect the context of the graduating university and the employer/organisation. While the layout and application of programs varied, programs generally consist of study days, time management, role adjustment, a preceptor/mentor, and work/life balance with hands-on-learning in a non-threatening environment and an opportunity for reflection [37].

However, the successful transition of new graduate nurses is multifactorial. To successfully transition a new graduate nurse, organisational culture such as a positive workplace culture is vital [39]. In the absence of other assistance, new graduate nurses rely on the broader nursing team. The value of the preceptor/mentor and the broader nursing team helped the new graduate nurses with a sense of belonging and lessened feelings of isolation [40]. Providing new graduate nurses with the support needed to perform their job is vital to their successful transition. New graduate nurses experience higher levels of psychological stress caused by work environments, which leads them to consistently seek employment elsewhere [41]. Collegial relationships with other staff members and feelings of teamwork solidify millennial nurses’ engagement and satisfaction in the work environment [42]. Preceptorship and mentorship improve confidence and reduce role stress in millennial nurses. According to Tan and Chin [42], millennial-generation nurses experience higher levels of burnout and emotional fatigue compared to other generations. Unlike previous generations, millennials “work to live”, and little to no engagement with the work environment will fuel the desire to search for a new job [41]. To effectively support new graduate nurses, whether they be millennials or from Generation Z, it is essential to understand their unique characteristics and needs. These generations, often described as tech-savvy, collaborative, and value-driven, require transition programs tailored to these traits to ensure their successful integration into the workforce. Consequently, retention strategies must focus on creating engaging and supportive work environments that align with their values and lifestyles.

Nursing has become increasingly complex, dynamic, and technologically advanced. New graduate nurses are expected to manage these challenges, even though some experienced nurses may have difficulty adapting to modern contemporary healthcare [43]. Nurses are expected to work in various specialised areas [44] and be flexible in adapting to contemporary healthcare practice [45]. Given the reported gap between theory and practice [24], along with the associated patient safety risks [26], there is a critical need to offer support to new graduate nurses in their initial year of practice. This support aims to foster relationships that enhance their capabilities in patient care, boost confidence and competence, and contribute to higher levels of job satisfaction and retention rates. New graduate nurses require the watchful eye and support of experienced nurses in an environment that will nourish their growth while buffering external pressures.

## 3. Recommendation

Addressing the needs and challenges of new graduate nurses requires a multifaceted approach that integrates support mechanisms tailored to their unique characteristics. This discussion paper outlines potential solutions to address the theory–practice gap and provides targeted support aimed at fostering a supportive work environment, enhancing professional development opportunities, and addressing specific generational traits. These measures are essential to ensure the successful transition of new graduates into the nursing workforce.

### 3.1. Structured Orientation and Training Sessions

The transition from nursing school to professional practice can be challenging for new graduate nurses. To ensure they are well-prepared to provide high-quality patient care, structured orientation and training programs are essential [46]. These initiatives help bridge the gap between academic learning and clinical application, fostering both competence and confidence [47].

Effective transition programs significantly enhance new graduate nurses’ confidence and reduce attrition rates. They should include structured sessions covering essential nursing skills, hospital protocols, and patient care standards [48]. Preceptorship and mentorship—highlighted in both the WHO Global Strategic Directions for Nursing and Midwifery 2021–2025 [49] and the State of World’s Nursing 2025 report [50]—are key components, with experienced nurses guiding new graduates through their first year, offering support, reducing stress, and reinforcing clinical confidence [37].

Hands-on learning in a supportive, low-pressure environment allows new graduates to refine their skills and receive constructive feedback. Regular reflection sessions provide opportunities to share experiences, address challenges, and celebrate achievements. By cultivating a structured and nurturing environment, these programs empower new graduate nurses to advocate for their patients and deliver safe, effective care [46].

Investing in the development of new graduate nurses not only supports their individual growth but also strengthens the overall quality and sustainability of the healthcare system.

### 3.2. Positive Organisational Culture

A positive organisational culture plays a pivotal role in the successful integration of new graduate nurses. Supportive leadership—characterized by attentiveness to individual needs—is essential. Fostering teamwork and collegial relationships helps create a sense of belonging and mitigate feelings of isolation [40].

Policies that promote work–life balance, such as flexible scheduling and adequate time off, are particularly important for new graduate nurses striving to manage both professional and personal lives [41]. A healthy workplace culture enhances job satisfaction, supports career development, and ensures new graduate nurses feel valued and recognised [51].

The WHO Global Strategic Directions for Nursing and Midwifery 2021–2025 [49] emphasises the importance of cultivating enabling environments that support professional growth, psychological safety, and leadership development. Creating a workplace culture where new graduate nurses feel valued, heard, and supported is essential for enhancing job satisfaction and reducing early attrition. This aligns with global calls for improved working conditions and leadership capacity as key drivers of nurse retention and performance. By fostering such a culture, healthcare organisations can not only strengthen staff retention but also elevate patient care standards and improve overall operational efficiency.

### 3.3. Support Services

Support services are critical in helping new graduate nurses navigate the demands of their roles while maintaining their well-being [46]. Younger nurses, in particular, are more susceptible to burnout and emotional fatigue [42]. Providing access to mental health resources—including counselling, stress management programs, and peer support groups—can make a significant difference.

These services offer safe spaces for new graduate nurses to express concerns, seek guidance, and build resilience. Regular breaks during shifts and wellness initiatives focused on physical and mental health—such as exercise, nutrition and mindfulness—further support overall well-being [46].

Calleja et al. [52], in their synthesis of 24 reviews, highlighted the potential of gratitude interventions to enhance resilience and reduce stress-related health impacts among new graduate nurses. Encouraging gratitude may help mitigate burnout and foster a more positive outlook.

The State of the World’s Nursing 2025 report [50] highlights the critical role of mental health and well-being in ensuring workforce sustainability, particularly in the aftermath of the COVID-19 pandemic. Supporting nurses’ emotional resilience is no longer optional—it is essential. Providing structured support systems aligns with the WHO Global Strategic Directions for Nursing and Midwifery [49], which emphasises the importance of service delivery and workforce protection. These systems ensure that nurses are not only clinically competent but also emotionally equipped to manage the demands of their roles. A robust support framework not only improves retention but also fosters a sense of value, belonging, and empowerment among new graduate nurses, enabling them to thrive both personally and professionally [46].

### 3.4. Continued Education Opportunities

New graduate nurses are eager to grow professionally and advance their career. Offering continued education is a powerful way for healthcare organisations to support this ambition [53]. As the healthcare landscape evolves, staying current with emerging practices, technologies, and treatments is essential [53]. Educational initiatives—such as workshops, certifications, and advance training—equip new graduate nurses with the skills and knowledge needed to deliver safe, high-quality care [53].

Beyond clinical competence, ongoing education fosters confidence, reduces burnout, and enhances job satisfaction. It also provides clear pathways for career progression through advanced degrees, specialty certifications, and leadership development programs [21]. When new graduate nurses see tangible opportunities for growth within their organisation, they are more likely to remain engaged and committed [53].

A culture of lifelong learning, which aligns with the WHO’s strategic direction on education [49], not only supports individual development but also strengthens the healthcare workforce. Investing in continued education ultimately benefits patients, nurses, and the broader healthcare system [54].

### 3.5. Integrating E-Learning Platforms

With the increasing integration of technology in healthcare, e-learning platforms have become a vital tool for nursing education and professional development. New graduate nurses, often well-versed in digital tools, are particularly well-positioned to benefit from these platforms [55].

E-learning offers flexible, accessible, and up-to-date learning opportunities that align with the fast-paced nature of healthcare [21]. These platforms support continuous learning through modules on clinical skills, emerging treatments, and regulatory updates [53]. Nurses can engage with the content at their own pace, making it easier to balance learning with the demands of their roles.

The WHO Global Strategic Directions for Nursing and Midwifery 2021–2025 [49] advocates for the integration of digital technologies into nursing education and practice to expand access, improve efficiency, and future-proof the workforce. E-learning platforms play a vital role in supporting the development of clinical competence, regulatory awareness, and adaptability—core competencies for navigating today’s complex healthcare environments. Moreover, digital learning fosters resilience and flexibility, enabling nurses to continuously update their knowledge and skills in response to evolving patient needs and technological advancements. As healthcare delivery becomes increasingly digitized, investing in robust e-learning infrastructure ensures that nurses remain confident, capable, and well-prepared to deliver safe, high-quality care in a rapidly changing landscape.

### 3.6. Finances

Financial pressures are a significant concern for new graduate nurses, impacting both recruitment and retention [21]. Many face student loan debt, high living costs, and salaries that may not reflect the demands of their roles [56]. These stressors can lead to job dissatisfaction, burnout, and even an early exit from the profession [57].

When financial concerns dominate, new graduate nurses may seek employment in higher-paying sectors or countries, take on excessive overtime, or relocate to more affordable areas—all of which contribute to high turnover rates and disrupt the continuity of care [56,57]. To address this, healthcare organisations must offer competitive salaries and comprehensive benefit packages. Financial incentives such as relocation assistance, childcare support, and loan repayment programs can significantly improve job satisfaction and retention [21].

It is also important to recognise the diversity within the new graduate cohort. While many are younger, others are mature-aged, balancing family responsibilities and financial obligations. Tailored financial and career support strategies that reflect these varied needs are essential for successful integration and long-term retention.

The State of the World’s Nursing 2025 report [50] underscores the importance of equitable compensation and targeted financial support mechanisms, particularly in high-cost urban areas and underserved regions. To address these challenges, healthcare organisations should offer competitive salaries, relocation assistance, childcare support, and student loan repayment programs. These measures not only alleviate financial stress but also demonstrate a commitment to supporting nurses across different life stages and circumstances.

By implementing inclusive and responsive financial strategies, healthcare systems can foster a more stable and engaged workforce, reduce turnover, and ensure that new graduate nurses—regardless of age or background—feel valued, supported, and empowered to build lasting careers in the profession

### 3.7. Work—Life Balance

Work–life balance is a top priority for younger generations of nurses, particularly millennials and Gen Z [42]. Unlike previous cohorts, they place greater emphasis on flexibility, mental health, and personal well-being [42,58].

To meet these expectations, healthcare organisations must adopt flexible staffing models. Options such as part-time roles, job sharing, self-scheduling, and remote work (where applicable) can help nurses better manage their professional and personal lives [59]. Roles in telehealth or administration, for example, offer opportunities for remote work while maintaining clinical engagement [42]. This aligns with the WHO’s call for workforce sustainability and retention strategies that reflect generational shifts and evolving workforce expectations [49].

Creating a supportive and inclusive workplace culture is equally important. This includes recognising staff contributions, fostering teamwork, and ensuring adequate staffing levels to prevent burnout [60]. Providing regular breaks, mental health resources, and opportunities for professional development further supports well-being [58].

A workplace that values work–life balance not only enhances job satisfaction but also strengthens retention and improves patient care outcomes [60].

## 4. Conclusions

It is imperative to acknowledge the substantial investment made in training new graduate nurses and the critical importance of retaining them within the workforce. The premature departure of these professionals not only results in an immediate loss of financial and educational resources but also diminishes the future capacity of healthcare services.

Retention efforts must therefore be proactive, responsive, and tailored, recognising the evolving demands of educational institutions, healthcare organisations, and societal expectations. Support mechanisms for new graduate nurses must be dynamic and individualised, taking into account the diversity of their educational pathways, personal circumstances, and career aspirations.

As the global demand for nurses continues to rise, there is a timely opportunity to deepen our understanding of the factors that promote retention, job satisfaction, and long-term engagement in the profession. Economic pressures, including mounting university loan debts and the rising cost of living—especially in metropolitan areas—underscore the need for healthcare systems to offer competitive and appealing career trajectories.

Younger nurses are increasingly seeking opportunities for professional growth, financial stability, and meaningful work. To remain competitive, healthcare organisations must adapt to these expectations and invest in strategies that support career development and well-being.

The diversification of healthcare delivery models presents new and exciting professional avenues for nurses. Leaders in healthcare must remain attuned to generational differences within the workforce and provide targeted guidance and support to new graduates. By fostering environments that nurture early-career nurses, we can reduce turnover, enhance professional fulfilment, and build a resilient nursing workforce equipped to meet the challenges of the future.

## Data Availability

This paper is an opinion piece authored by practicing clinicians who are also engaged in academic roles at large metropolitan hospitals. It draws upon the extensive clinical and research experience of the authors. As such, the insights and perspectives presented are grounded in professional expertise rather than primary data collection. The relevant literature has been cited throughout the paper to support and contextualize the viewpoints expressed. No new datasets were generated or analysed for this study.

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
