# Peer review of "How Do We Keep Our New Graduate Nurses in Australia?"

_nursrep, 2025, doi:10.3390/nursrep15080276_

Round 1
Reviewer 1 Report
Comments and Suggestions for Authors
Thank you for the opportunity to read this very enjoyable opinion paper. The paper gathers many of the multiple factors contributing to the nursing workforce shortage to provide a summary from which several recommendations were provided. Researchers globally would be able to weave local challenges and facilitators into the recommendations provided in this paper to prioritize and implement strategies suitable for their context and new graduate nurses.
My edits are very minor and likely would be picked up in the page proofs but I will offer them in the hope that they are helpful.
P 3 of 14: possible typo in the word “process” which maybe should read “possess” in the sentence; “Both Millennial and Gen Z nurses process similar professional skills learning and communication styles and outlook reflecting their exposure to current technology and social trends.”
P 3 of 14: possible typo(s) in the sentence; “Lack of preparedness in the transition of new graduate nurses and attributed to the theory-practice gap, has been extensively documented in the literature as persisting with minimal change for several decades [24].”
Reviewer 2 Report
Comments and Suggestions for Authors
Dear authors
Thank you such a thoughtful paper on a an issue that needs critical attention. Attached please find some inputs that could further strengthen your paper.

Author Response
Please see feedback attached

Reviewer 3 Report
Comments and Suggestions for Authors
Dear author, I hope this message finds you well. Thank you for submitting your paper and for the opportunity to review the article “How do we keep our new graduate nurses?”. After a comprehensive review, some areas for improvement were identified. The introduction should provide the structure of the article to better inform the reader. The conclusion is not sufficiently developed. Authors should state possible limitations of the research, future directions of the research, and properly emphasize the practical and theoretical implications of the paper. Authors should distinguish more consistently between the author’s experience and empirical evidence. It is necessary for authors to unify their terminology, as they use different terms such as “new graduate nurse”, “novice nurse”, “early career nurse”, “entry-level nurse”. Support is also mentioned in the article, in some parts it should be specified what type of support is involved. The methodological part of the work is completely absent from the article. Although this is a discussion article, the authors claim to be based on a synthesis of literature and professional experience, but no systematic approach to selecting or evaluating sources is described. On what basis were they selected? It is necessary to clearly distinguish between ambiguities between opinions and clear evidence from the literature. There is excessive repetition of information in certain parts of the article. Authors should avoid unspecified or generalized statements if they are not supported by anything.
Best regards.
Author Response
Please see attached report

Round 2
Reviewer 3 Report
Comments and Suggestions for Authors
Dear author,
I hope this message finds you well. Thank you for incorporating the comments. The comment "The introduction should provide the structure of the article to better inform the reader." was meant to mean that you should add a paragraph in the introduction where you tell the reader what chapters your scientific article consists of and briefly describe them.
Best regards.
Author Response
Please see attached feedback
